# Solvent-Assisted Adsorption of Cellulose on a Carbon Catalyst as a Pretreatment Method for Hydrolysis to Glucose

**Abhijit Shrotri [1,*], Kiko Eguchi [1], Lina Mahardiani [1,2], Hirokazu Kobayashi [1,3], Masakuni Yamashita [4], Hiroshi Yagita [5] and Atsushi Fukuoka [1,*]**

[1] Institute for Catalysis, Hokkaido University, Kita 21 Nishi 10, Kita-ku, Sapporo 001-0021, Japan
[2] Faculty of Teacher Training and Education, Sebelas Maret University, Jl. Ir. Sutami 36A, Kentingan, Surakarta 57126, Indonesia
[3] Komaba Institute for Science, Graduate School of Arts and Sciences, The University of Tokyo, 3-8-1 Komaba, Meguro-ku, Tokyo 153-8902, Japan
[4] ETISA Co., Ltd., 2-17-11-204 Ryogoku, Sumida-ku, Tokyo 130-0026, Japan
[5] Nippon Institute of Technology, 4-1 Gakuendai, Miyashiro-machi, Saitama 345-8501, Japan
[*] Correspondence: ashrotri@cat.hokudai.ac.jp (A.S.); fukuoka@cat.hokudai.ac.jp (A.F.); Tel.: +81-11-706-9137 (A.S.); +81-11-706-9140 (A.F.)

**Abstract:** Cellulose hydrolysis to glucose using a heterogeneous catalyst is a necessary step in producing bio-based chemicals and polymers. The requirement for energy-intensive pretreatments, such as ball milling, to increase the reactivity of cellulose is one of the major issues in this area. Here, we show that by using solvent-assisted adsorption as a pretreatment step, cellulose can be adsorbed on the surface of a carbon catalyst. For adsorption pretreatment, phosphoric acid ($H_3PO_4$) performed better than other solvents such as sulfuric acid ($H_2SO_4$), tetrabutylammonium fluoride/dimethyl sulfoxide (TBAF/DMSO) and 1-butyl-3-methylimidazolium chloride ([BMMI]Cl). Hydrolysis after the adsorption of cellulose and the removal of $H_3PO_4$ produced a 73% yield of glucose. Partial hydrolysis of cellulose in $H_3PO_4$ before adsorption increased the final glucose yield. The glucose yield was proportional to the number of weakly acidic functional groups on the carbon catalyst, indicating the reaction was heterogeneously catalyzed. In a preliminary lab-scale life-cycle analysis (LCA), greenhouse gas (GHG) emissions per kg of glucose produced through the hydrolysis of cellulose were calculated. The $H_3PO_4$-assisted adsorption notably reduces GHG emissions compared to the previously reported ball milling pretreatment.

**Keywords:** cellulose; hydrolysis; glucose; carbon catalyst; phosphoric acid

## 1. Introduction

The depolymerization of cellulose through hydrolysis or hydrolytic hydrogenation in an aqueous phase is one of the most important reactions in biomass conversion (Figure 1) [1–3]. Hydrolysis produces glucose, which is a precursor to 5-hydroxymethylfurfural (5-HMF) [4]. On the other hand, the direct hydrolytic hydrogenation of cellulose produces sorbitol [5,6], which is used to produce isosorbide [7]. 5-HMF is on the verge of being commercialized for use in the synthesis of polyethylene 2,5-furandicarboxylate [8], and isosorbide is already being used commercially for the synthesis of polycarbonates [9]. While the technology required for the synthesis of renewable polymers from these platform chemicals is maturing, the depolymerization of cellulose through hydrolysis or hydrolytic hydrogenation is still not feasible.

The stagnation of research concerning cellulose depolymerization is a result of the high cost of the pretreatments required for its processing [10]. A pretreatment is used to separate cellulose from lignocellulosic biomass in a process called fractionation [11]. Pretreatment is also used to increase the reactivity of cellulose by reducing the degree

of crystallinity [12,13]. Even if recycled paper is used as a source of cellulose to avoid fractionation, pretreatment is still required to increase its reactivity [14].

**Figure 1.** Depolymerization of cellulose by hydrolysis to glucose and further hydrogenation to sorbitol.

The reactivity of native cellulose is low because of its insoluble and crystalline nature that inhibits chemical and biological attacks. The crystalline structure of cellulose prevents access to the glycosidic bonds of cellulose molecules and may also cause steric hindrance for the transformation of the tetrahydropyran ring in cellulose to a half-chair conformation, which is necessary for hydrolysis [15]. Therefore, the objective of an effective pretreatment step is to break the crystalline structure of cellulose in order to increase the permeability of chemical species. Such a pretreatment step converts crystalline cellulose to its amorphous form, which tends to be more reactive.

Historically, concentrated mineral acids such as $H_2SO_4$ and HCl were used for the hydrolysis of cellulose [16,17]. These processes did not require pretreatment because of their ability to dissolve cellulose by breaking the hydrogen bond network. However, the processes suffered from issues of corrosion, the separation of acid and waste disposal. The use of solid acid catalysts eliminates these drawbacks associated with mineral acids. Solid acid catalysts can be easily recycled, and they do not cause corrosion. Among various solid acid catalysts, porous carbon materials with acidic functional groups are particularly active in terms of cellulose hydrolysis [3]. In comparison to metal oxides or acidic resins, carbon catalysts preferentially adsorb cellulose molecules on the surface and promote the rate of hydrolysis [18–20]. Carbon catalysts with sulfonated or oxygenated functional groups have consistently shown high activity toward cellulose hydrolysis [21–24]. Despite these advantages, the use of solid carbon catalysts is not yet practical, owing to the requirement for cellulose pretreatment to increase its reactivity. In addition, when solid acid catalysts are used, pretreatment also has an important role in increasing the interaction between the carbon catalyst and cellulose.

Ball milling is a conventional pretreatment step that changes the crystalline form of cellulose to an amorphous form. Amorphous cellulose, which has a disordered structure, tends to be more reactive than its crystalline counterpart, but it still has limited interaction with solid catalysts owing to its insoluble nature. However, the rate of cellulose hydrolysis can be increased by an order if the cellulose molecules are adsorbed on the surface of the catalyst prior to or during the reaction [25]. The adsorption of soluble β-1,4-cello-oligosaccharides on the carbon surface is instantaneous and thermodynamically favored [18,20,26]. However, the insolubility of cellulose molecules larger than decamer having 10 monomer units in conventional solvents prevents their adsorption on the surface of the carbon.

Mix-milling is an effective strategy to pretreat and adsorb cellulose on the surface of a carbon catalyst by the combined solid-state milling of a carbon catalyst and cellulose. After this pretreatment, the rate of cellulose conversion increased 13-fold, and a high yield of soluble products was obtained at 180 °C for 20 min [27]. However, the high energy requirement is a major drawback of using ball milling pretreatment [28]. Therefore, it is essential to develop economical methods for the adsorption of cellulose on carbon surfaces to achieve a high rate of hydrolysis.

Here, we investigate the solvent-assisted adsorption of cellulose on carbon catalysts as a pretreatment step for the heterogeneously catalyzed hydrolysis of cellulose. The dissolution of cellulose in various solvents, including $H_3PO_4$, has been reported as a means of producing amorphous cellulose, which is more active toward enzymatic hydrolysis [29–32]. In this study, we have used various solvents to dissolve cellulose and then adsorb them on a carbon catalyst by means of adding an antisolvent, followed by its hydrolysis under mild conditions to obtain glucose. The carbon catalyst was prepared through the oxidation of activated carbon in air to introduce weakly acidic functional groups.

## 2. Materials and Methods

Microcrystalline cellulose (Avicel PH-101) was purchased from Sigma Aldrich. Activated carbon (denoted as AC) was supplied by Ajinomoto Fine-Techno (product name BA). Activated carbon Norit was purchased from Sigma-Aldrich. Sulfuric acid (98%), phosphoric acid (85%), dimethyl sulfoxide (DMSO) and 1-butyl-3-methylimidazolium chloride ([BMIM]Cl) were purchased from Wako Fujifilm Chemicals, and tetrabutylammonium fluoride trihydrate (TBAF) was purchased from Tokyo Chemical Industry.

The X-ray diffraction (XRD) pattern of cellulose was measured with a Rigaku Ultima IV instrument using a CuK$\alpha$ X-ray ($\lambda$ = 1.54 Å) operating at 40 kV and 20 mA. A scanning electron microscope (SEM) image was taken using a JEOL JSM 7400F microscope. The sample was placed on the stage using double-sided copper tape and coated with a thin layer of Pt before analysis.

The oxidation of carbon materials to produce air-oxidized carbon catalysts was performed in the presence of air [33,34]. Activated carbon AC or Norit (4.0 g) was spread on a Pyrex dish with a diameter of 130 mm and a uniform thickness. The sample was then heated in an electric furnace under air with the following program: 25 °C to 120 °C at a rate of 10 °C min$^{-1}$ and then maintained at 120 °C for 2 h to remove the physisorbed water, followed by heating to 425 °C at a rate of 4 °C min$^{-1}$ and then maintained at 425 °C for a further 10 h. The obtained oxidized carbon catalysts were denoted as AC-Air or Norit-Air.

The solvent-assisted adsorption of cellulose was performed by dissolving cellulose in various solvents and then precipitating them in the presence of a carbon catalyst by using an antisolvent. For each experiment, 0.324 g of microcrystalline cellulose (corresponding to 2 mmol of glucose units) was placed in a round-bottomed flask, and 0.9 mL of water was added to wet the cellulose particles. Then, 16.2 mL of solvent (typically cold $H_3PO_4$ (85%)) was added, and the slurry was stirred for 15 min. A homogeneous solution was obtained after some time. To perform partial hydrolysis, the solution was heated to 60 °C and stirred for the required time. Then, the required amount of carbon catalyst (typically 0.324 g) was added, and the mixture was stirred at room temperature for an additional 10 min. Then, 30 mL of antisolvent was added (typically water), and the mixture was shaken vigorously. The solid-containing catalyst and adsorbed cellulose were separated by filtration and washed with 300 mL of water to remove excess $H_3PO_4$.

For a typical hydrolysis experiment, the wet solid obtained after solvent-assisted adsorption was transferred to a stainless-steel autoclave (OM Lab-Tech MMJ-100) with the help of 40 mL of water. The autoclave was sealed and heated to 180 °C (heating time about 11 min) and stirred (600 rpm). After maintaining the temperature at 180 °C for 20 min, the heater was removed, and the autoclave was cooled by an air blower. The reaction mixture was centrifuged to separate the solid residue and liquid product. The liquid product was analyzed using HPLC (Shodex Sugar SH-1011 column and Phenomenex Rezex RPM-Monosaccharide Pb$^{++}$ column). All of the products were quantified through absolute calibration using standard compounds. Cellobiose and cellotriose were obtained as oligomers, and these were also quantified through absolute calibration with standard compounds. The residual solid was dried at 110 °C overnight and then weighed to calculate cellulose conversion.

The number of acidic functional groups on the carbon catalysts was calculated by neutralizing the carbon using NaOH and then determining the NaOH consumed [35]. For

each analysis, 0.5 g of carbon was added to 20 mL of NaOH solution (0.05 M). The resulting dispersion was stirred for 24 h under Ar at 25 °C. Then, the solid and liquid were separated by filtration, and 5 mL of HCl solution (0.05 M) was added to 5 mL of the filtrate obtained after neutralization with NaOH. The resulting solution was back-titrated using NaOH solution (0.05 M) to calculate the consumption of the base in the first step. Two drops of methyl orange were added before titration to determine the endpoint. The amount of base consumed in the first neutralization step was attributed to the presence of the total number of acid sites.

Based on the results of the hydrolysis experiments, the greenhouse gas (GHG) emissions associated with the production of glucose from cellulose were estimated using a simple life cycle assessment (LCA) method. Cellulose was assumed to be the raw material, and the chemicals (e.g., phosphoric acid) used in the glucose production process were also included. The amount of electricity used in the glucose production process was estimated based on the catalogs of the thermostatic chambers and others. The GHG emissions from wastewater treatment in the production process were also included in the calculation using the LCA database [36]. In addition, for the $H_3PO_4$ used in a series of processes that can be reused, the recycling rate was set to 90–99%. The conditions for the dissolution of cellulose in $H_3PO_4$ were assumed to be two cases: (1) 25 °C for seven days and (2) 60 °C for 60 min. It was also assumed that the carbon catalyst was completely recycled.

## 3. Results and Discussion

The hydrolysis of cellulose using a heterogeneous catalyst requires efficient contact between cellulose molecules and carbon particles [27]. The cellulose used in this study (Avicel PH-101) was crystalline in nature and was made up of irregular particles with a size of 100 μm (Figure 2). The carbon catalyst was made up of elongated particles with a size range of 50–100 μm. A detailed characterization of the AC-Air catalyst used in this study was described in our previous reports [26,37]. The AC-Air catalyst was found to be the most active for the hydrolysis of cellulose after ball milling pretreatment.

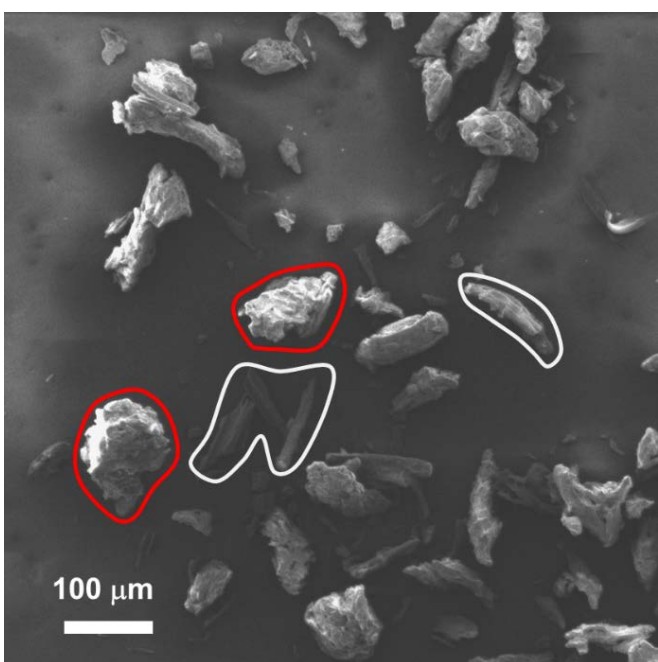

**Figure 2.** SEM image of a physical mixture of cellulose and AC-Air carbon catalyst. Cellulose particles are encircled in red, and AC-Air particles are encircled in white.

Initially, we tested the hydrolysis of untreated crystalline cellulose in a batch reactor at 180 °C for 20 min in the presence of water. The yield of glucose was only 0.1%, with a

cellulose conversion of 2.3% after hydrolysis when no catalyst was used. In the presence of AC-Air, a carbon catalyst containing 2.64 mmol g$^{-1}$ acidic functional groups, the yield of glucose increased to 2.6% (along with 1.2% fructose), and the conversion increased to 12%. The concentration of both cellulose and catalyst in the reaction mixture was 8.1 mg mL$^{-1}$ each. Under these conditions, the interparticle collision between cellulose and carbon governs the rate of hydrolysis. Considering the large particle size of cellulose and the carbon catalyst observed in the SEM image, it can be safely assumed that interparticle interaction was limited. Nevertheless, the presence of a catalyst had a positive effect on the conversion of cellulose despite the limited interaction between the cellulose and carbon particles.

Next, we investigated solvent-assisted adsorption to enhance the proximity of cellulose molecules and the active sites on the catalyst. A scheme of the complete adsorption and hydrolysis method is shown in Figure 3. In a typical method, cellulose was dissolved in the solvent, and then the carbon catalyst was added to the mixture. This was followed by the addition of water, which acts as an antisolvent to promote the adsorption of cellulose on the catalyst surface. After further washing with water, the cellulose–catalyst composite was hydrolyzed in the presence of water.

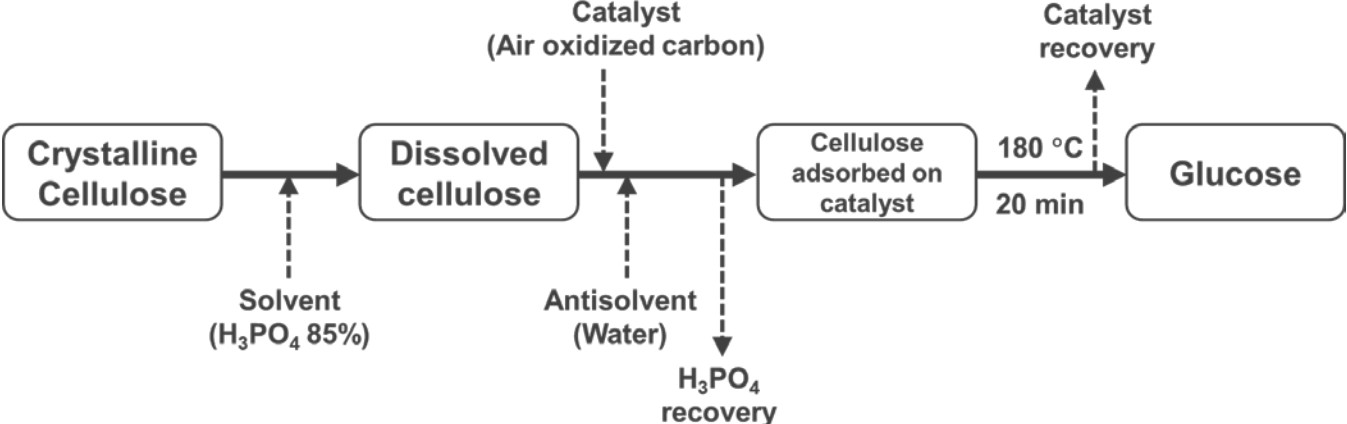

**Figure 3.** Schematic representation of solvent-assisted adsorption process followed by hydrolysis of adsorbed cellulose to obtain glucose.

Cellulose dissolution in phosphoric acid was performed by adding 85% H$_3$PO$_4$ to wet cellulose and stirring for 15 min at room temperature. Sulfuric acid (72%), tetra-butylammonium fluoride (TBAF) in dimethyl sulfoxide (DMSO) (40 wt.% solution) and 1-butyl-3-methylimidazolium chloride ([BMIM]Cl) were also used as solvents, and cellulose was completely dissolved in all of these solvents. The dissolved cellulose was precipitated by adding water as an antisolvent. In order to adsorb cellulose on the carbon surface, a carbon catalyst was added before the addition of water. The catalyst–cellulose composite was filtered and washed with more water and then used for hydrolysis. The hydrolysis of the cellulose–AC–Air mixture obtained after the dissolution pretreatment produced 16% glucose when H$_3$PO$_4$ was used as a solvent (Figure 4). A small amount of oligomers and byproducts, such as fructose, 5-HMF and levoglucosan, were also formed. Unidentified products were also obtained at a 17% yield. The H$_2$SO$_4$ pretreatment produced glucose at a 12% yield. Glucose was not obtained when the TBAF/DMSO solution was used as a solvent, and no other products were detected. The complete deactivation of the acidic sites on the carbon surface by the fluoride ion of TBAF was the likely cause of the poor yield. [BMIM]Cl was a poor solvent for the adsorption of cellulose as glucose was not produced, and some unknown compounds were formed.

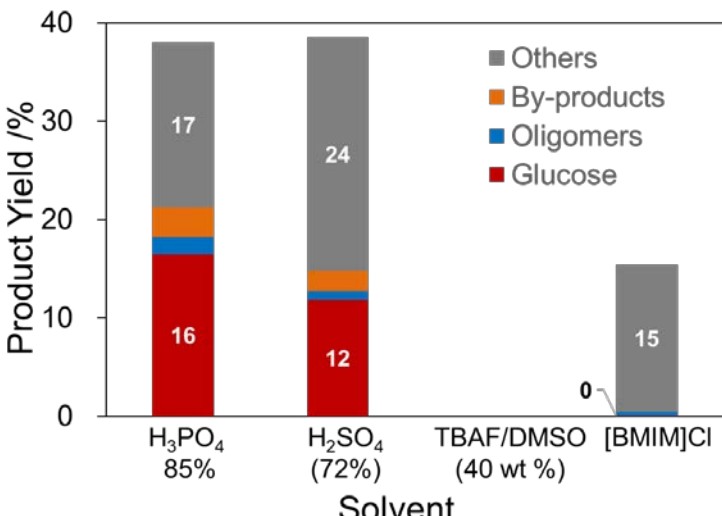

**Figure 4.** Yield of products after hydrolysis of cellulose treated with different solvents and precipitated in the presence of AC–Air catalyst. Cellulose treatment condition: 0.324 g cellulose, 16.2 g solvent, 15 min stirring, 0.324 g AC–Air, precipitation with 30 mL $H_2O$ followed by washing with 300 mL $H_2O$. Hydrolysis reaction conditions: wet cellulose + AC-Air mixture obtained after pretreatment, 40 mL water, 180 °C, 20 min. By-products include fructose, 5-HMF and levoglucosan. "Others" is the difference between cellulose conversion and identifiable products.

$H_3PO_4$ (85%) was chosen as the preferred solvent for further study owing to its better performance and a lower tendency to corrode equipment in comparison to $H_2SO_4$ (72%). The cellulose dissolved in $H_3PO_4$ was precipitated in the absence of a carbon catalyst and analyzed using X-ray diffraction. The diffraction pattern resembled the native structure of cellulose I (Figure 5, red line). Therefore, molecular dissolution was not complete, even though the solution apparently looked clear. Heating the cellulose/$H_3PO_4$ solution to 60 °C for 30 min ensured the complete dissolution of the cellulose, as evidenced by the X-ray diffraction of the precipitated cellulose, which showed the characteristic peaks of a regenerated cellulose II structure (Figure 4, black line) [31]. When the cellulose was precipitated in the presence of a carbon catalyst after the heating treatment, the intensity of the XRD peaks of the cellulose II structure diminished (Figure 4, blue line), indicating that the presence of AC-Air inhibited the recrystallization of cellulose.

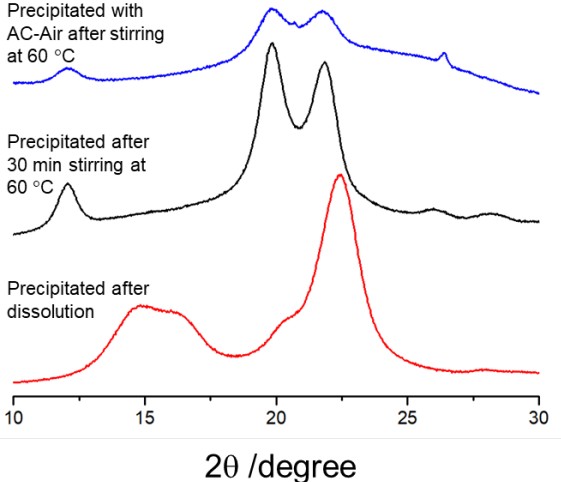

**Figure 5.** X-Ray diffraction patterns of cellulose precipitate after dissolution in $H_3PO_4$ under different conditions.

The heating treatment of cellulose to ensure complete dissolution also improved the glucose yield. The hydrolysis of cellulose adsorbed on AC-Air after a 60 °C treatment for 30 min resulted in a glucose yield of 66% (Figure 6a). Further increasing the heating time to 60 min produced a glucose yield of 73%. The glucose yield decreased to 57% when the heating time was further increased to 90 min. Heating the cellulose solution in the presence of 85% $H_3PO_4$ caused the partial hydrolysis of cellulose molecules, reducing their degree of polymerization and enhancing their adsorption on the carbon surface. If the partial hydrolysis time is extended, smaller oligosaccharides may be formed that will not precipitate when the antisolvent is added. The slow partial hydrolysis of cellulose occurred even at room temperature (25 °C). Stirring the cellulose dissolved in $H_3PO_4$ for up to seven days at room temperature increased the glucose yield to 70% (Figure 6b). Even after the partial hydrolysis of cellulose, its adsorption on carbon was necessary. In a control experiment, cellulose was precipitated in the absence of AC-Air after 6 days of treatment at room temperature, and then the hydrolysis reaction was performed in the presence of externally added AC-Air at 180 °C 20 min. The glucose yield reduced to 36% in comparison to 62% when precipitation was performed in the presence of carbon. This result attests to the importance of the adsorption of dissolved cellulose on the catalyst surface.

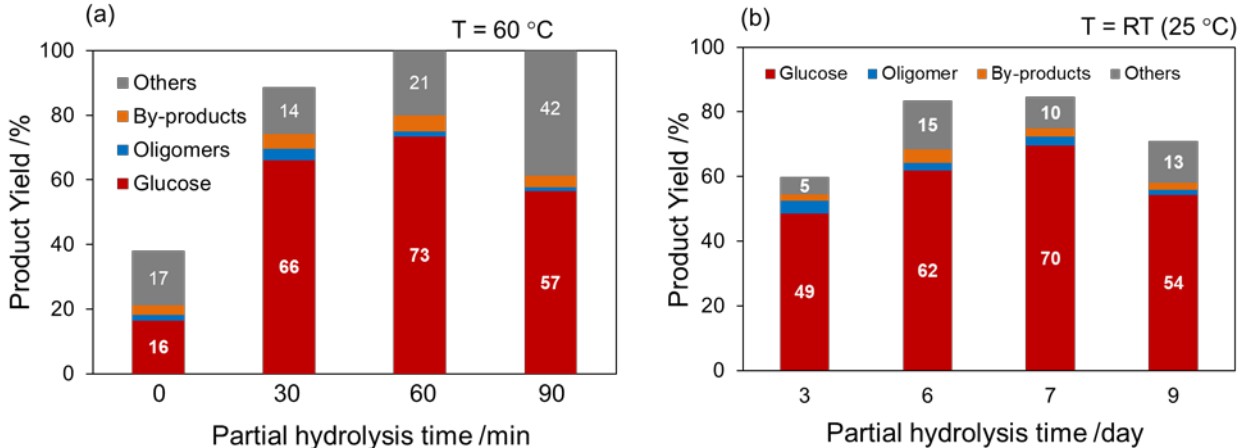

**Figure 6.** Product yield after hydrolysis of cellulose dissolved in $H_3PO_4$ and hydrolyzed partially for various times at (**a**) 60 °C and (**b**) room temperature before adsorption on AC-Air. Reaction conditions: Wet cellulose + AC-Air mixture, 40 mL water, 180 °C, 20 min. By-products include fructose, 5-HMF and levoglucosan. "Others" is the difference between cellulose conversion and identifiable products.

The role of the carbon catalysts in terms of glucose yield was also evaluated. In comparison to AC-Air, which produced a 73% yield of glucose, the performance of unoxidized AC was poor, resulting in only 9% glucose (Figure 7a). This result was not much better than the reaction in the absence of any catalyst, which produced only 4% glucose. Therefore, the presence of oxygenated functional groups was crucial for obtaining a high glucose yield. Using another type of activated carbon called Norit, the importance of oxygenated functional groups was clearly established. The glucose yield with unoxidized Norit was 12%, which increased to 58% in the presence of air-oxidized Norit-Air. The total number of acidic functional groups on all four catalysts was calculated by using titration with NaOH. The yield of glucose was correlated to the total number of acidic functional groups on the catalyst surface (Figure 7b). Our group previously reported a similar correlation between weakly acidic functional groups with a yield of glucose when cellulose was pretreated with ball milling [3]. Therefore, with respect to the hydrolysis of cellulose in the presence of a carbon catalyst, solvent-assisted adsorption is a good pretreatment method for glucose synthesis.

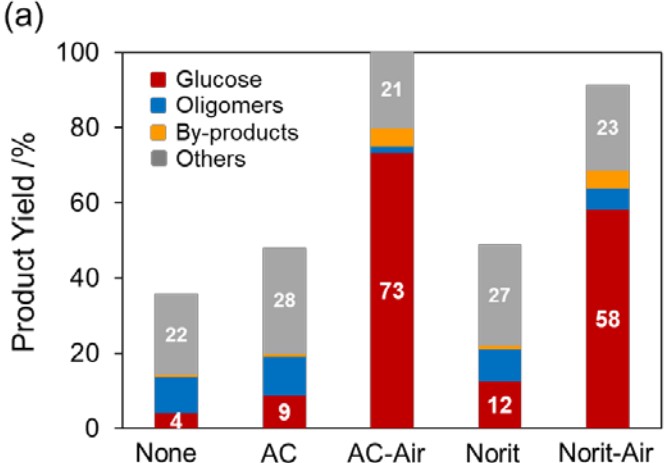
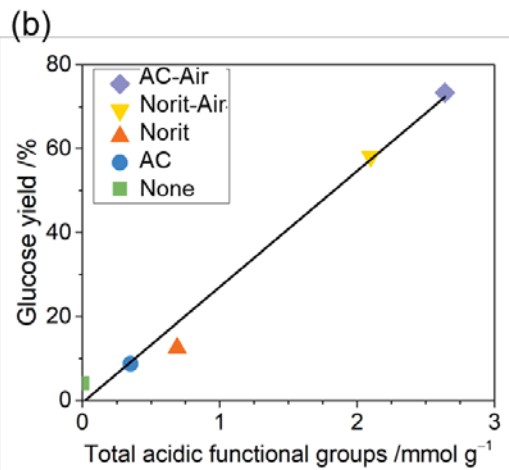

**Figure 7.** (**a**) Yield of products after hydrolysis of pretreated cellulose adsorbed on various catalysts. (**b**) Correlation between glucose yield and the number of acidic functional groups present on each catalyst. Cellulose treatment conditions: 0.324 g cellulose, 16.2 g $H_3PO_4$, 15 min stirring at room temperature, then 60 min stirring at 60 °C, 0.324 g AC-Air, precipitation with 30 mL $H_2O$ followed by washing with 300 mL $H_2O$. Hydrolysis reaction conditions: wet cellulose + catalyst mixture, 40 mL water, 180 °C, 20 min. By-products include fructose, 5-HMF and levoglucosan. "Others" is the difference between cellulose conversion and identifiable products.

It should be noted that the amount of oligomer produced was low in all cases. In our previous study, a small amount of mineral acid, such as 0.012 wt% HCl or $H_3PO_4$, was necessary to accelerate the hydrolysis of soluble oligomers to glucose [25,27,38]. The pH of the reaction solution (wet AC-Air and cellulose composite with 40 mL $H_2O$) in this study was less than 3.0, indicating the presence of residual $H_3PO_4$ in the solution, which would catalyze the conversion of the oligomer to glucose. In addition, the further reaction of glucose to products such as levoglucosan and 5-HMF was limited, which was beneficial to obtain higher glucose yields [39].

Altering the amount of catalyst had a slight influence on the catalytic activity. At a high catalyst loading (substrate to catalyst ratio S/C = 0.5), glucose yield decreased to 66%, and more unknown products were formed (Figure 8). At S/C of 2 and 3, the glucose yield decreased, and a slight increase in the formation of oligomers was observed, indicating the incomplete hydrolysis of the adsorbed oligomers. Therefore, in our study, an S/C of 1 was optimal to achieve a maximum glucose yield of 73%.

The antisolvent used to adsorb the cellulose on the carbon catalyst can influence the adsorption process. We evaluated the use of different antisolvents during the adsorption and recovery of $H_3PO_4$. In comparison to water, solvents such as ethanol, acetone and 1-propanol were used. On the one hand, the use of organic solvents makes the process less green, but in terms of reconcentrating $H_3PO_4$, the use of an organic solvent can reduce the energy demand. The use of organic solvents had a detrimental effect on glucose yield after hydrolysis, as shown in Figure 9a, and glucose yields of 58–61% were obtained. The presence of organic solvents is likely to hinder the adsorption of cellulose because the adsorption process is driven by CH-π and hydrophobic interaction, which are promoted in aqueous environments [18]. In addition, a comparison of the X-ray diffraction patterns of cellulose after the $H_3PO_4$ treatment and precipitation in the presence of different antisolvents showed that the formation of a cellulose II crystal phase was beneficial for higher glucose yields in the presence of water and 1-propanol (Figure 9b).

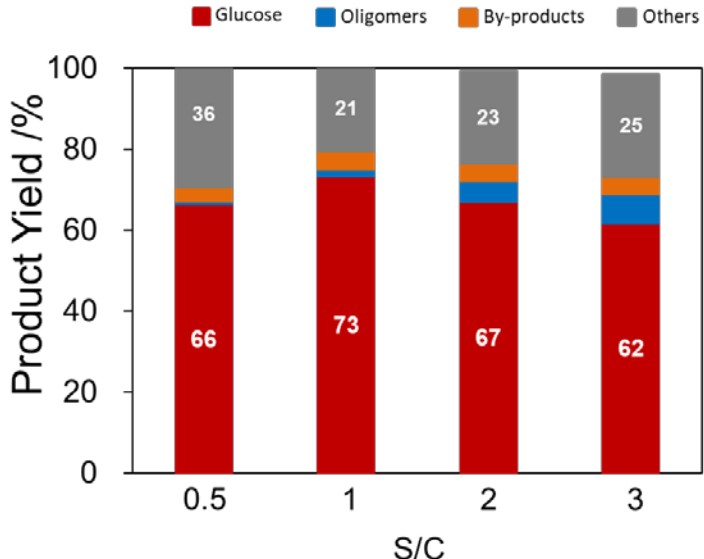

**Figure 8.** Yield of products after hydrolysis of cellulose pretreated with $H_3PO_3$ with different substrate-to-catalyst ratios. Wet cellulose + AC-Air mixture, 40 mL water, 180 °C, 20 min. By-products include fructose, 5-HMF and levoglucosan. "Others" is the difference between cellulose conversion and identifiable products.

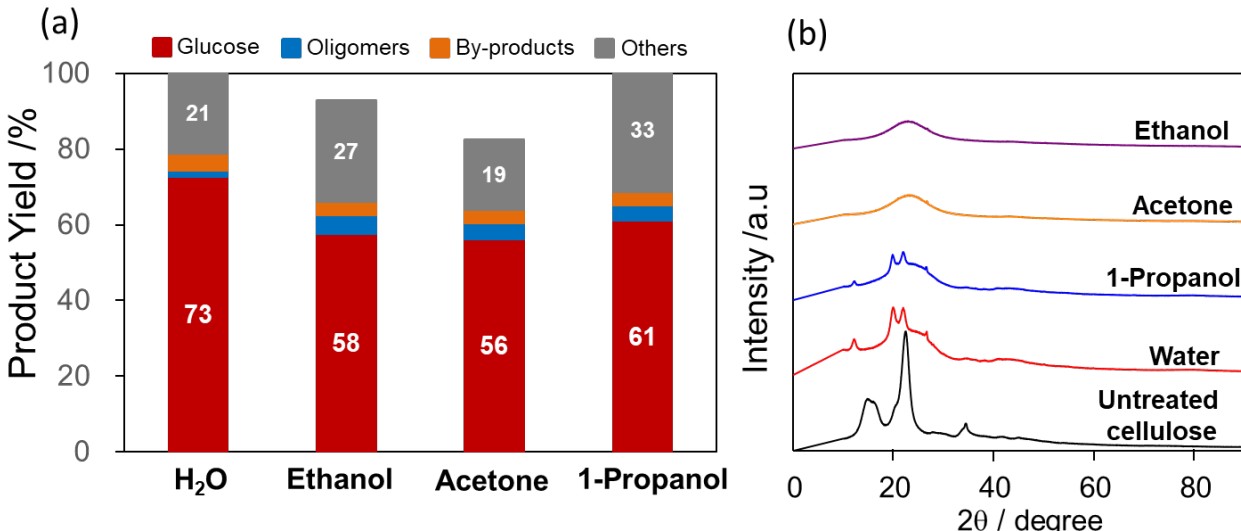

**Figure 9.** (**a**) Influence of antisolvent during $H_3PO_4$ treatment of cellulose during hydrolysis reaction. (**b**) X-Ray diffraction patterns for cellulose precipitated in different antisolvent after dissolution in $H_3PO_4$.

We performed a preliminary LCA to assess the feasibility of the $H_3PO_4$ pretreatment under two conditions: (1) the dissolution of cellulose and partial hydrolysis at 25 °C for seven days and (2) the dissolution of cellulose and partial hydrolysis at 60 °C for 1 h. Glucose yield was set at 73% for these two cases. The detailed conditions are summarized in the Supplementary Materials. Figure 10 shows the GHG emissions for cases (1) and (2) by assuming a 90–99% recovery of $H_3PO_4$. In addition, we have also compared the LCA analysis of cellulose hydrolysis after the mix-milling pretreatment of cellulose and a carbon catalyst, as reported by us previously [26,34]. Mix-milling was conducted using a ball mill at 25 °C for 2 h, and a 90% total yield of glucose and oligomers was considered. For this case (1), GHG emissions per kg of glucose produced based on an $H_3PO_4$ recovery of 90%, 95% and 99% were calculated as 20.5, 13.5 and 7.8 kg-$CO_2$eq, respectively (Figure 10). The

GHG emissions were slightly increased in case (2) and were calculated as 22.6, 15.8 and 10.2 kg-$CO_2$eq, respectively. In the case of the mix-milling pretreatment of cellulose and the carbon catalyst, the GHG emissions were calculated as 42.0 kg-$CO_2$eq. Therefore, the $H_3PO_4$-assisted adsorption approach can indeed reduce GHG emissions in comparison to the ball milling pretreatment. For example, if we use the partial hydrolysis condition at 60 °C for 1 h, the GHG emissions would be reduced to a quarter, with 99% recycle of the $H_3PO_4$ used.

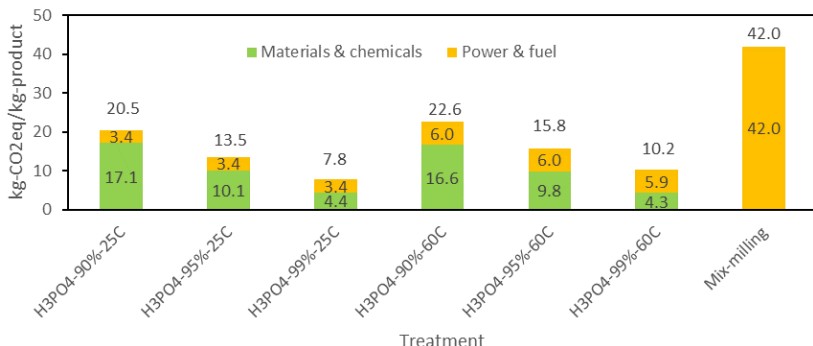

**Figure 10.** $CO_2$ emissions per kg of glucose produced after hydrolysis of cellulose pretreated with $H_3PO_4$-assisted adsorption method and mix-milling method. Hydrolysis conditions: cellulose + AC-Air mixture, 40 mL water, 180 °C, 20 min.

These GHG emissions were calculated using LCA methods based on the laboratory-scale-level results, and more accurate LCA can be performed if a specific process design is determined. In addition, recent progress has been made in designing production facilities that maximize the use of renewable energy sources, such as solar and wind power generation, and the introduction of renewable energy sources can put an even lower-carbon manufacturing process in perspective.

## 4. Conclusions

We have shown that the solvent-assisted adsorption of cellulose over a carbon catalyst is an effective way to pretreat cellulose for the purpose of its hydrolysis using a heterogeneous catalyst. In our results, 85% $H_3PO_4$ was the most suitable solvent for dissolving and adsorbing cellulose over a carbon catalyst in comparison to $H_2SO_4$ (72%), (TBAF/DMSO) and [BMMI]Cl ionic liquid. The hydrolysis of adsorbed cellulose in the presence of AC-Air, a catalyst containing a high density of oxygenated functional groups, resulted in a glucose yield of 73%. The partial hydrolysis of cellulose in $H_3PO_4$ at 60 °C for 60 min was required to increase the adsorption of cellulose on the surface of the carbon catalyst. The yield of glucose was linearly correlated with the number of acidic functional groups on the catalyst surface, indicating that the influence of residual $H_3PO_4$ was negligible toward hydrolysis activity. LCA analysis showed that at the lab scale, the $H_3PO_4$-assisted adsorption significantly reduces the GHG emissions in comparison to the previously reported ball milling pretreatment method.

**Supplementary Materials:** The following supporting information can be downloaded at: https://www.mdpi.com/article/10.3390/chemistry5010028/s1. The Supplementary Materials contain detailed procedures and conditions for LCA for the phosphoric acid treatment method and the mix-milling methods.

**Author Contributions:** Conceptualization, A.S., H.K., M.Y., H.Y. and A.F.; methodology, A.S., H.K., M.Y. and H.Y.; validation, A.S. and M.Y.; investigation, A.S, K.E. and L.M.; writing—original draft preparation, A.S.; writing—review and editing, A.S., H.K., M.Y. and A.F.; visualization, A.S. and M.Y.; supervision, A.F.; project administration, A.F.; funding acquisition, A.S. and A.F. All authors have read and agreed to the published version of the manuscript.

**Funding:** This research was funded by the Japan Science and Technology Agency (JST) ALCA (JPMJAL1309) and JSPS KAKENHI Grant Number JP21H01708. This work was partially supported by the Yanmar Environmental Sustainability Support Association (KI0222043).

**Data Availability Statement:** Not applicable.

**Conflicts of Interest:** The authors declare no conflict of interest.

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
