# Peer review of "Solvent-Assisted Adsorption of Cellulose on a Carbon Catalyst as a Pretreatment Method for Hydrolysis to Glucose"

_chemistry, doi:10.3390/chemistry5010028_

Round 1

Reviewer 1 Report

Overall, a very nicely written and executed study. However, I feel it needs some corrections/changes before publication. Please see my comments directly in the attached PDF file.

Author Response

Point 1:  5-hydroxymethyl furfural  à 5-hydroxymethylfurfural

Response 1: Changed as per suggestion.

Point 2: These conditions are not mild.

Response 2: We have removed the phrase suggesting the conditions are mild.

Point 3: What's up with this weird number 0.324 g?? How about 0.25 or 0.5 g?

Response 3: We chose 0.324 g of cellulose because it corresponds to about 2 mmol of glucose monomer units based on the molecular weight of cellulose (162.14 g/mol). We have clarified this point in the experimental section of the manuscript. In addition, we have consistently used this value in our previous study which facilitates comparison of data. Ref [25,34] in the manuscript.

Point 4: How this temperature was chosen? Based on your previous work?

Response 4: Yes, the temperature was chosen based on our previous work using the same carbon catalyst with ball milling pretreatment. Ref [25,34] in the manuscript.

Point 5: Did you mix the solution with hand as with H3PO4 and other solvent it can form a very viscous solution.

Response 5: The solution was mixed using magnetic stirrer. None of the solutions were viscous enough to inhibit stirring.

Point 6: Did you run control without catalyst? Also how many times did you wash it?

Response 6: Yes, we did control reaction without catalyst. This result is already shown in Fig 7a.

For the washing step we used 300 mL of water. This information is also present in the experimental section.

Point 7: I would also give these treatment conditions?

Response 7: We have modified the figure caption to include treatment conditions for pretreatment.

Point 8: DId you try giving it longer time at room temperature? Check this paper out: 1. Percival Zhang YH, Cui J, Lynd LR, Kuang LR. A transition from cellulose swelling to cellulose dissolution by o-phosphoric acid: Evidence from enzymatic hydrolysis and supramolecular structure. Biomacromolecules. 2006;7:644–8.

Response 8: Yes, we did partial hydrolysis for longer time at room temperature. The results of that experiment are already present in manuscript (See Fig 6b). The above mentioned paper by Zhang et al. infers that when H3PO4 with more than 80% concentration is used the solution turns transparent within a few minutes. We have used 85% H3PO4 in our experiment and visually the solution became transparent in our case too. However, additional analysis showed that cellulose I structure was preserved indicating incomplete dissolution.

Point 9: But how do we know if H2SO4 at 60C is not going to prove to be a better solvent?

Response 9: It is true that upon partial hydrolysis in the presence of H2SO4 the glucose yield from cellulose might increase. However, in our initial screening of solvent (Fig 4), pretreatment in H2SO4 produced more unknown products (24%) in comparison to H3PO4. This implies that the stronger acidity of H2SO4 was causing side reaction and this effect would be exacerbated upon treatment at 60 °C. Therefore, further optimization with H2SO4 was not performed. Moreover, H2SO4 is also not an effective choice owing to its corrosive properties. Therefore, it was not considered for further study.

Point 10: It is little surprising to see that cellulose II can be prepared in the presence of H3PO4 because usually it requires some exotic conditions such concentrated NaOH?

  1. Mittal A, Katahira R, Himmel ME, Johnson DK. Effects of alkaline or liquid-ammonia treatment on crystalline cellulose: Changes in crystalline structure and effects on enzymatic digestibility. Biotechnol Biofuels. 2011;4.

Response 10: Contrary to the opinion of the reviewer, formation of cellulose II does not require exotic condition. Dissolution of cellulose in H3PO4 and precipitation in aqueous systems results in rearrangement of the crystal structure to the thermodynamically stable cellulose II form. This phenomenon has been documented earlier by other researchers.

Jia, Xuejuan, et al. "Preparation and characterization of cellulose regenerated from phosphoric acid." Journal of Agricultural and Food Chemistry 61.50 (2013): 12405-12414.

Kassem, Ihsane, et al. "Phosphoric acid-mediated green preparation of regenerated cellulose spheres and their use for all-cellulose cross-linked superabsorbent hydrogels." International journal of biological macromolecules 162 (2020): 136-149.

Point 11: I am not sure what is the basis for these random conditions? How about using lower or higher temp?

Response 11: In our experiment the degree of partial hydrolysis during pretreatment was a function of treatment temperature and time. As we show in Fig 6b, treatment at room temperature required up to 7 days to achieve maximum glucose yield of 70%. Similar result was obtained at 60 °C and a reasonable treatment time of 60 min. Indeed increasing or decreasing the treatment temperature would change the time required to achieve maximum glucose yield. However, such optimization would not provide any further insight into the process and would not add to the scientific discussion in the paper.

Point 12: AGain, I would provide cellulose treatment conditions here as well to avoid any confusion. In M&M section, the method to qualtify oligomers also needs to be discussed.

Response 12: We have added the cellulose treatment condition to the caption of Fig. 7 and have also added the method to quantify oligomers in the experimental section (Line 137-138).

Point 13: What was the cellulose conversion for all these conditions?

Response 13: In Figure 8 cellulose conversion for all reactions was close to 100 %. As mentioned in the figure caption, The “others” (grey portion of bar graph) was calculated by subtracting the cellulose conversion from the yield of identified products.

Point 14: Did you look at a little higher/lower hydrolysis and/or cellulose treatment temp.? Also, it would be great to include some data if you had only added the catalyst during hydrolysis stage? I meant without adsorption?

Response 14: In our previous study we reported that 180 °C and 20 min was the optimum condition for cellulose hydrolysis after combined mix-milling. To facilitate the comparison with our previous work we only investigated these conditions.

We did a control reaction by adding the catalyst during hydrolysis stage. The glucose yield reduced to 36% when catalyst was added during hydrolysis in comparison to 62% when the catalyst was added before precipitation of cellulose (partial hydrolysis of 6 days at room temperature was applied for both reactions). Therefore, It was evident that adsorption of cellulose over carbon catalyst during precipitation was essential for high hydrolysis activity. We have added this explanation to the revised manuscript (Line 242 – 248).

Point 15: Also it is a good idea to include some data on the recyclebility of the catalyst.

Response 15: We have previously reported the recyclability of the AC-Air catalyst therefore we did not include similar data to avoid repetition.

Shrotri, A., et al. "Cellulose hydrolysis using oxidized carbon catalyst in a plug-flow slurry process." Industrial & Engineering Chemistry Research 56.49 (2017): 14471-14478.

Point 16: But these spectra are extaclty like what you had with carbon catalyst which you argued that carbon catalyst inhibited recryst. If this is the case then with ethanol and acetone, the yield shoould be higher? Clarify.

Response 16: In addition to inhibition of crystallization it is also crucial that the adsorption of cellulose over carbon happens. Because the adsorption over carbon catalyst happens through CH-p and hydrophobic interaction the presence of organic solvent would hinder the adsorption. Therefore, we propose that the primary influence in reduction of activity is due to limited adsorption. We have modified the discussion to this effect (Line 302 – 304).

Reviewer 2 Report

Authors present an interesting work about the conversion of cellulose into glucose by means of activated carbon catalysts using phosphoric acid as impregnation solvent, over other pre-treatment methods, as ball or mix-milling developed by the same research group. However, I believe that some aspects of the work should be clarified or corrected:

1Phosphoric acid treatment of biomass is a common practice to obtain chemicals. However, authors do not give any reference about that.

TThe number of acidic functional groups on AC-Air catalysts was calculated by titration with NaOH (133-142 lines), and that correlated with the yield of glucose after hydrolysis step (243-245 lines, as well as figure 7b). Authors do not explain why they get a glucose yield as high as 73% using as hydrolytic catalyst theWet cellulose + AC-Air mixture’ after 60 min partial hydrolysis step. Regarding their own results (i.e. references 21 and 26) trace amounts of acid (0.012 wt % HCl) were necessary to obtain near complete hydrolysis of cellulose. Similarly, I think that the solid ‘Wet cellulose + AC-Air mixture’, used as catalyst during the 180ºC/20 min step, may contain trace amounts of H3PO4 retained on AC-acidic functional groups, as non-innocent actor of the cellulose hydrolysis to glucose by air oxidized AC.  Therefore, I believe that solid ‘Wet cellulose + AC-Air mixture’ should be also titrated after its water washing and before its use as hydrolytic catalyst. 

TThe air-oxidized carbon, used by authors at 50 wt % as ‘catalyst’, was recovered, in an unspecified amount, after hydrolytic process, and reused. The amount of recovered catalyst must be considered in the LCA calculation, because the proportion 1:1 (wt/wt) is very high and the preparation of air-oxidized AC requires too much energy.

Author Response

Authors present an interesting work about the conversion of cellulose into glucose by means of activated carbon catalysts using phosphoric acid as impregnation solvent, over other pre-treatment methods, as ball or mix-milling developed by the same research group. However, I believe that some aspects of the work should be clarified or corrected:

Point 1: Phosphoric acid treatment of biomass is a common practice to obtain chemicals. However, authors do not give any reference about that.

Response 1: We have added several reference about phosphoric acid treatment of biomass and cellulose (Line 91-93).

Point 2: The number of acidic functional groups on AC-Air catalysts was calculated by titration with NaOH (133-142 lines), and that correlated with the yield of glucose after hydrolysis step (243-245 lines, as well as figure 7b). Authors do not explain why they get a glucose yield as high as 73% using as hydrolytic catalyst the ‘Wet cellulose + AC-Air mixture’ after 60 min partial hydrolysis step. Regarding their own results (i.e. references 21and 26) trace amounts of acid (0.012 wt % HCl) were necessary to obtain near complete hydrolysis of cellulose. Similarly, I think that the solid ‘Wet cellulose + AC-Air mixture’, used as catalyst during the 180ºC/20 min step, may contain trace amounts of H3PO4 retained on AC-acidic functional groups, as non-innocent actor of the cellulose hydrolysis to glucose by air oxidized AC. Therefore, I believe that solid ‘Wet cellulose + AC-Air mixture’ should be also titrated after its water washing and before its use as hydrolytic catalyst.

Response 2: We agree with the reviewer’s opinion that trace amount of phosphoric acid might catalyze the conversion of oligomers to glucose. However, titration of wet carbon + cellulose mixture would not give accurate results due to presence of acid sites on carbon catalyst. Instead, we found that the pH of solution before hydrolysis reaction was less than 3.0 in all cases. This indicated that residual amount of H3PO4 was present in the solution, which would catalyze the hydrolysis of oligomers to glucose. We have included this discussion in the manuscript (Line 269-274).

Point 3: The air-oxidized carbon, used by authors at 50 wt % as ‘catalyst’, was recovered, in an unspecified amount, after hydrolytic process, and reused. The amount of recovered catalyst must be considered in the LCA calculation, because the proportion 1:1 (wt/wt) is very high and the preparation of air-oxidized AC requires too much energy.

Response 3: Indeed, the preparation of AC-Air is an energy consuming process. In the LCA calculation, we have already considered 100 % recycle of AC-air. We have modified the experimental section to clarify this fact (Line 159-160).

Reviewer 3 Report

This article is devoted to the topic of catalytic hydrolysis of cellulose on a carbon catalyst. The authors investigated the adsorption of cellulose with a solvent on a catalyst for processing the preparation of a substrate for hydrolysis. On the one hand, the topic of cellulose hydrolysis has been studied for a long time and a lot of data on this issue has been accumulated in science, on the other hand, interest in this topic has not waned, and new catalysts and conditions for the hydrolysis of cellulose to low molecular weight products are currently being studied. In other words, this article is in the trend of modern research. The article makes a good impression, but there are some points that need to be improved:

1. Abstract can be expanded.

2. It is desirable to indicate more clearly why the authors chose this particular catalyst.

3. In the introduction, it is desirable to indicate examples of the use of carbon catalysts for hydrolysis. In particular, cite: 10.1039/C7GC02143G, 10.1016/j.biortech.2012.03.098, 10.1007/s42452-019-1776-6, 10.1021/ja803983h and others.

4. Throughout the article, when describing the data, it is desirable to provide more comparison with the literature data. This will give more validity to the conclusions.

5. Please cite and compare your data with work: 10.1007/s00226-022-01363-4.

6. It is desirable to indicate more clearly in the text what other products are formed by the authors as a result of hydrolysis on activated carbon.

7. Conclusions can also be expanded and made more concise.

Author Response

This article is devoted to the topic of catalytic hydrolysis of cellulose on a carbon catalyst. The authors investigated the adsorption of cellulose with a solvent on a catalyst for processing the preparation of a substrate for hydrolysis. On the one hand, the topic of cellulose hydrolysis has been studied for a long time and a lot of data on this issue has been accumulated in science, on the other hand, interest in this topic has not waned, and new catalysts and conditions for the hydrolysis of cellulose to low molecular weight products are currently being studied. In other words, this article is in the trend of modern research. The article makes a good impression, but there are some points that need to be improved:

Point 1: Abstract can be expanded.

Response 1: We have expanded the abstract as per the request to include additional details.

Point 2: It is desirable to indicate more clearly why the authors chose this particular catalyst.

Response 2: We have previously reported the synthesis and use of this catalyst for hydrolysis of cellulose after ball milling pretreatment. Therefore, for comparison on pretreatment methods we chose to use the same catalyst (Line 167-168).

Point 3: In the introduction, it is desirable to indicate examples of the use of carbon catalysts for hydrolysis. In particular, cite:10.1039/C7GC02143G, 10.1016/j.biortech.2012.03.098,10.1007/s42452-019-1776-6, 10.1021/ja803983h and others.

Response 3: We have added relevant citations in the introduction and have expanded the discussion (Line 68–69).

Point 4: Throughout the article, when describing the data, it is desirable to provide more comparison with the literature data. This will give more validity to the conclusions.

Response 4: We have added additional comparison of data with literature as per the request.

Point 5: Please cite and compare your data with work:10.1007/s00226-022-01363-4.

Response 5: We have cited the above-mentioned literature (Line 274 -276).

Point 6: It is desirable to indicate more clearly in the text what other products are formed by the authors as a result of hydrolysis on activated carbon.

Response 6: We have added a description of the by-products and other products obtained during analysis (Line 202–204).

Point 7: Conclusions can also be expanded and made more concise.

Response 7: We have expanded the conclusion.

Round 2

Reviewer 1 Report

The revised manuscript looks good and can be accepted as is.

Reviewer 3 Report

accepted